# MalHAPGNN: An Enhanced Call Graph-Based Malware Detection Framework Using Hierarchical Attention Pooling Graph Neural Network

**DOI:** 10.3390/s25020374

**Published:** 2025-01-10

**Authors:** Wenjie Guo, Wenbiao Du, Xiuqi Yang, Jingfeng Xue, Yong Wang, Weijie Han, Jingjing Hu

**Affiliations:** 1School of Computer Science and Technology, Beijing Institute of Technology, Beijing 100811, China; wenjieguo@bit.edu.cn (W.G.); duwenbiao@bit.edu.cn (W.D.); xqyang@bit.edu.cn (X.Y.); xuejf@bit.edu.cn (J.X.); wangyong@bit.edu.cn (Y.W.); 2School of Space Information, Space Engineering University, Beijing 100084, China; sec_hwj2006@hgd.edu.cn

**Keywords:** malware detection, malware embedding, graph neural network, representation learning, graph pooling mechanism

## Abstract

While deep learning techniques have been extensively employed in malware detection, there is a notable challenge in effectively embedding malware features. Current neural network methods primarily capture superficial characteristics, lacking in-depth semantic exploration of functions and failing to preserve structural information at the file level. Motivated by the aforementioned challenges, this paper introduces MalHAPGNN, a novel framework for malware detection that leverages a hierarchical attention pooling graph neural network based on enhanced call graphs. Firstly, to ensure semantic richness, a Bidirectional Encoder Representations from Transformers-based (BERT) attribute-enhanced function embedding method is proposed for the extraction of node attributes in the function call graph. Subsequently, this work designs a hierarchical graph neural network that integrates attention mechanisms and pooling operations, complemented by function node sampling and structural learning strategies. This framework delivers a comprehensive profile of malicious code across semantic, syntactic, and structural dimensions. Extensive experiments conducted on the Kaggle and VirusShare datasets have demonstrated that the proposed framework outperforms other graph neural network (GNN)-based malware detection methods.

## 1. Introduction

Malware detection is a pivotal aspect of software security, focused on identifying and mitigating malware threats to computer systems and networks. According to AV-TEST [1], 122,322,459 new malware samples were generated in the past 12 months, indicating that future cyber attacks will likely become more frequent and severe. As malware becomes increasingly complex, traditional signature-based detection methods are no longer sufficient to counter new threats. In response, numerous studies have applied machine learning and deep learning to malware detection and classification. These methods are primarily divided into static analysis and dynamic analysis. Static analysis does not execute the program but extracts features by analyzing the program’s source code or binary files. In contrast, dynamic analysis requires executing the program and monitoring its runtime behavior, typically conducted in a sandbox or virtual environment to capture detailed interactions between the program and the system. Static analysis has become the foundation of many malware detection systems due to its efficiency.

Traditional signature-based static detection methods have been proven inadequate, prompting the development of machine learning [2] and deep learning [3] approaches that reduce reliance on expert knowledge. Particularly, neural networks can automatically extract higher-level abstract features, enabling end-to-end model training and prediction [4]. Pioneering works such as code2vec [5] and SAFE [6] have demonstrated that embeddings can convert code into a continuous vector format, facilitating neural network processing and subsequent classification. In addition, researchers have explored extracting rich information from code to detect malware, including syntax, semantics, and structure [7,8,9]. And various methods such as converting binary files into images [10], extracting control flow graphs (CFGs) [11], and constructing function call graphs [12] have been investigated.

For the purpose of file-level malware detection and classification, function call graphs are employed to strike a delicate balance between the depth of semantic information and the economic efficiency of the preprocessing stage [12]. The graphical representation of these functional relationships offers a systematic portrayal of the program’s architecture, which is indispensable for discerning the operational nuances of malware. In the pursuit of enhanced file-level malware embedding with graph data, researchers have primarily focused on two key aspects: extracting attributes of functions within call graphs and effectively utilizing these graphs for file-level embedding. In terms of function attribute extraction, existing embedding methods may encounter difficulties in accurately capturing semantic information of functions, especially when dealing with codebases that contain a multitude of instructions and are inherently complex. Furthermore, graph convolutional networks (GCNs) [13,14] offer a novel perspective and methodology for effectively integrating the embedded representations of different functions to represent the entirety of a malicious code program.

Nevertheless, current malware detection methods suffer from two major limitations. (i) Function-level embeddings suffer from incomplete semantic capture. Existing methods either ignore this structural information by treating the entire instruction as one word, such as InnerEye [15] and EKLAVYA [16]. Specifically, some approaches use control flow graphs (CFGs) to capture contextual information between instructions [17]. However, due to evolutionary techniques such as code instruction reordering, the context information on control flow could be noisy and does not reflect the actual dependencies between instructions. In addition, function-level embedding is not currently applied to malware detection tasks, such as Gemini [18] for binary code similarity detection, EKLAVYA [16] for function type signature inference, and deepvsa [19] for value set analysis. DeepSemantic [20] constructs a semantically sensitive code representation method based on BERT for benign files, and provides a data preprocessing method to solve the out of vocabulary (OOV) problem. The OOV problem refers to the issue where certain words or tokens appear in the test data but were not present in the training data. This can lead to the model’s inability to recognize and process these new terms, thereby affecting its performance. In the context of malware detection, the OOV problem may result in the model’s failure to accurately identify and analyze new variants of malware or those employing unknown techniques, thus reducing the accuracy and reliability of detection. (ii) Graph neural networks have issues with indistinguishable node feature updates and a lack of exploration of structural semantics. Within the framework of GCNs, nodes optimize their feature representations by integrating information from their neighborhoods. Although GCNs can capture the local neighborhood information of nodes, their standard implementations typically treat all nodes within the neighborhood as equal-weight contributors, such as MalwareExpert [12]. In malware analysis, there are irrelevant codes introduced through obfuscation techniques, which do not affect the execution flow of the program, but interfere with the updating of the node feature representation. Therefore, in order to accurately reflect the role of nodes in malware behavior, the influence of nodes in the neighborhood must be differentiated.

To tackle the aforementioned challenges, in this paper, we propose a novel framework, namely MalHAPGNN, with two key components, BERT-based enhanced function call graph construction and a hierarchical attention pooling graph neural network (HAPGNN). Specifically, we introduce normalization methods for disassembled functions and improve pre-training tasks. The proposed BERT-based function embedding (BBFE) method harnesses the sophisticated language processing capabilities of BERT to produce initial embeddings for functions, effectively encapsulating semantic information. HAPGNN employs attention mechanisms to hierarchically aggregate nodes in the call graph, preserving structural information and highlighting key nodes and subgraphs, thus enabling comprehensive malware representation for graph-level classification. In particular, HAPGNN integrates contextual graph information to calculate node significance, ensuring rational node and subgraph sampling. And the pooling process employs structural learning to maintain subgraph connectivity, ensuring sufficient semantic transmission. The entire learning process can be optimized end-to-end.

In summary, our main contributions are as follows:

1. This paper introduces a novel BBFE method that effectively addresses the OOV issue while retaining semantic information.

2. This work proposes an innovative HAPGNN, incorporating a comprehensive node sampling method and a graph structure learning mechanism. HAPGNN supports deep semantic mining and enables a thorough and in-depth representation of malware.

3. This research conducts experiments on two widely used public datasets, Kaggle and VirusShare. The results demonstrate the effectiveness of the proposed method and its superiority over a range of other GNN-based malware detection techniques.

The rest of this paper is organized as follows: Section 2 reviews related work. Section 3 presents the problem formulation. Section 4 details the methodology, Section 5 discusses the experiments, and Section 6 concludes the paper.

## 2. Related Work

### 2.1. Static Malware Detection

The fixed-length vector of a binary file based on feature engineering is the key to using machine learning models to detect malware. Some researchers have carried out manual analyses of malicious code, and then proposed hand-designed features for detection. Anderson et al. proposed EMBER [7], an ML-based malware detection model, which extracted PE structure, byte entropy, string, and many static analysis features, and then classified them by lightGBM. Priyanka Singh [8] proposed a static malware detection method based on feedforward deep neural networks (FFDNNs). Although this approach has achieved some success by extracting features from executable header information, it mainly relies on hand-extracted features, limiting its ability to detect more complex or mutated malware.

The use of embeddings as features is often used in natural language processing. Embeddings can be used to hold contextual information. Many research efforts have proposed methods for generating embeddings from sequential or structured data [21,22,23]. There are also embeddings for the explicit design of source code and program binaries, such as code2vec [5] and SAFE [6]. In order to further extract semantic information, malicious code detection based on function call graphs is also a common method. By extracting graph structure features, function features, and other information, the malicious code is detected by machine learning or rule matching. In [24], the authors propose SeGDroid to realize the recognition of malware by learning the semantic knowledge of a sensitive function call graph (FCG). This method uses the graph convolutional neural network algorithm to generate the graph embedding of the sensitive FCG, and then uses the machine learning algorithm to train the malware detection model.

MalwareExpert [12] uses function call graphs, combined with Asm2Vec and SAFE graph embedding techniques, using a graph convolutional network (GCN)-based model, which can be used to process complex APT (Advanced Persistent Threat) samples. It shows excellent detection ability and accuracy, and provides interpretability. Zhang et al. [25] propose a spectrum-based directed graph network (SDGNet) approach to solve the difficult problem of processing directed graphs in malware detection. Traditional spectral-based graph neural networks are difficult to apply to digraphs because of the asymmetry of adjacency matrix of digraphs. SDGNet uses three weighted graph matrix normalization methods to transform the graph adjacency matrix into three symmetric graph matrices describing different aspects of node information interaction, which effectively processes the directed graph data and improves the accuracy of malware classification. In addition, SDGNet uses multi-level multifaceted directed GCN (MDGCN) extraction graph representation, and combines the graph representation of different levels to form comprehensive classification features, which further enhances the detection performance.

MalGraph [26] proposes a hierarchical graph neural network, which combines function call graphs and control flow graphs to construct hierarchical graph representation, learns global graph vectors through a graph neural network and pool network, and realizes Windows malware detection. MAGIC [11] uses a depth map convolutional neural network for classifying malware represented by control flow graphs. The approach focuses on using graph structures to improve the accuracy of malware classification.

Above all, the use of graph neural networks for malware detection based on function call graphs has great potential, but there are also several points we need to pay attention to. On the one hand, the finer granularity of graph data extraction and input necessitates a corresponding increase in resource consumption during the data preprocessing phase. Consequently, advanced data mining methods are essential for the comprehensive and efficient extraction of the intrinsic features of functions. On the other hand, the detection model needs to have stronger information mining ability and learn as much as possible the syntactic, semantic, and structural information in the graph data. Specifically, we will improve the data-driven function feature extraction method and construct the enhanced function attribute call graph. And we will modify the graph neural network model applicable to the specific scenario of malware detection to achieve more efficient and accurate graph embedding.

### 2.2. Graph Pooling Mechanism

The pooling operation of traditional convolutional neural networks can reduce the size of input feature maps and the number of training parameters. Similarly, the graph pooling layer is an indispensable part of graph embedding, and the computational complexity can be reduced, and overfitting problems can be avoided by generating simplified subgraphs. However, the pooling operator in CNNs cannot be directly applied to the irregular graph structure data, and significant efforts have been made to solve this problem.

The existing graph pooling methods are divided into global pooling and hierarchical pooling. Global pooling treats the input data as flat and regular structured data, and all nodes are treated equally, which tends to lead to the loss of rich structural information in the graph, while hierarchical pooling solves this problem. Hierarchical pooling solves the problem by simplifying the graph structure in a staged manner. At each level, it reduces the number of nodes to construct a higher-level representation of the graph. This approach allows the model to meticulously capture and retain important structural information at each step. Its advantages include the following: preserving structural information, which enables the retention of crucial graph structures and local details while reducing dimensions; and adaptability to complex tasks, making it suitable for tasks that require a deep understanding of graph structural features.

DiffPool [27] proposes to use neural networks to soft-cluster nodes to form a dense cluster allocation matrix, which is costly to compute. gPool [28] and SAGPool [29] design the top-K node selection process to form the induced subgraph of the next input layer. EdgePool [30] designs pooling operations by shrinking edges in the graph, but it is less flexible and always pools about half of the nodes. Set2Set [31] implements the global pooling operation by aggregating information with Long Short-Term Memory (LSTM) [32]. DGCNN [33] pools the graph according to the last channel of the feature map value, which is sorted in descending order. Pooling operations based on graph topology have also been discussed in the literature [34,35]. Within these works, methods such as Graclus [36] and graph coarsening techniques [37] are employed as pooling modules.

The above methods are summarized in Table 1. The aforementioned studies, whether employing global pooling methods or hierarchical pooling approaches, have primarily utilized local graph contextual information, neglecting the collaboration and contribution from different sources of information. MVPool [38], an extension of HGP-SL [39], introduces a multi-view pooling operation that ranks nodes across different views with varying contextual graph information. It also employs an attention mechanism to facilitate collaboration between different views, thereby generating robust node rankings.

Inspired by MVPool [38], this section posits the need for a graph pooling operation suitable for malware detection. Based on the aforementioned analysis, we propose that the pooling component can be enhanced in two specific respects. First, concerning the state update within layers, graph attention algorithms may be particularly well suited to scenarios in malicious code detection where functions of greater significance within the call graph require heightened attention. Second, with respect to the propagation of information between layers, there is a necessity for an improved method of calculating Node Importance Value for function nodes in the context of malicious code detection, which would facilitate more efficient sampling. Such an operation should enable comprehensive and in-depth mining of node information and a method for subgraph structure reconstruction. This would allow the proposed method to acquire broader and more comprehensive knowledge to support downstream graph embedding tasks.

## 3. Problem Formulation

Before delving into the specifics of the proposed malware detection architecture, this section begins with an analysis of the problem domain and the corresponding notation to facilitate a precise description in subsequent sections. This foundational approach ensures that the theoretical underpinnings and the mathematical formalisms are clearly laid out, providing a solid basis for understanding the complexities of the model and its application to malware detection.

Given any arbitrary malicious sample $M_{i}$, upon binary disassembly, preliminary sequences of assembly code and function sequences can be obtained. Each function is subjected to one-hot encoding, and the function call relationships are extracted. Specifically, $G_{i}=\left(V_{i},A_{i},H_{i}\right)$ is used to represent $M_{i}$, where the vertex set $V_{i}=\left\{v_{1i},v_{2i},\ldots ,v_{ni}\right\}$ represents the function nodes in the graph, and $n_{i}$ denotes the number of nodes contained in $G_{i}$, which is the number of functions in the malicious code. $A_{i}\in R^{n_{i}\times n_{i}}$ is the adjacency matrix of $G_{i}$, representing the function call relationships within the malicious code.

After enhanced function attribute extraction, once preliminary embedding vectors h for all nodes are obtained, the feature matrix $H_{i}\in R^{n_{i}\times f}$ for $G_{i}$ is derived, where *f* represents the feature dimension of the constructed enhanced function attributes. Furthermore, the model proposed in this paper includes a pooling process, during which the scale of the graph changes due to the reduction in the number of nodes. Therefore, this paper further uses $A_{i}^{k}$ and $H_{i}^{k}$ to represent the adjacency matrix and feature matrix of the *k*-th layer, with the total number of stacked layers denoted by *K*. The label of the malicious sample $M_{i}$ is denoted as $Y_{i}=\left\{0,1,\ldots ,C\right\}$, where for binary classification, C=1; for multi-class classification, *C* represents the number of categories. If $M_{i}$ belongs to category *j*, then $Y_{ij}=1$; otherwise, it is 0.

With the above notation, we formally define our problem as follows: given a dataset $D=\{G_{1},G_{2},\ldots ,G_{n}\}$ and label information $Y=\{Y_{1},Y_{2},\ldots ,Y_{n}\}$, with the given parameters—the number of stacked layers *K*, the pooling rate *r*, and the graph embedding layer dimension nhid—this work ultimately aims to train a model that can predict the category of malware samples.

## 4. Methodology

### 4.1. Overview

The overall architecture of the proposed methodology is depicted in Figure 1, which is primarily composed of three distinct parts. The first part is data preprocessing, where binary code is disassembled to extract assembly code sequences. With thorough data cleaning and function call graph extraction, assembly function sequences and function call relationships are obtained. The second part is the construction of enhanced attribute function call graphs based on a modified BERT. The BBFE technique is introduced following an in-depth analysis of the syntactic, semantic, and structural characteristics of malware, aimed at deriving enhanced function attributes. The final part is the embedding of the entire malware via a HAPGNN network for classification or detection. HAPGNN utilizes a hierarchical structure, where each set of graph attention and pooling layers is followed by a readout function that produces a graph representation at that level. These representations from various levels are then integrated to form the final graph embedding, which is subsequently processed by a multilayer perceptron (MLP) to output classification results. Specific introductions are provided in following sections.

### 4.2. Data Preprocessing

This section focuses on the rigorous process of data normalization and standardization to create an optimized corpus for function-level embedding, effectively mitigating the OOV problem and minimizing training expenses.

Employing BERT necessitates the construction of a vocabulary, which serves as the corpus. The embedding of functions may encounter the OOV issue, where rare words, derived terms, or compound words absent from the vocabulary cannot be converted into vectors for model input. The objective of this study is to condense superfluous semantic information and construct a compact vector space, thereby reducing the number of parameters and enhancing training efficiency.

Our investigation has identified a significant body of work dedicated to this field. Notably, DeepSemantic [20] focused on the normalization and standardization of operands, while MalOSDF [40] conducted semantic analysis of opcodes. However, DeepSemantic is tailored to function-level tasks, and MalOSDF constructs feature vectors based solely on opcodes for malicious code detection, which results in previous methods being overly fine-grained. Based on previous work, this study employs standardization rules depicted in Table 2 and Table 3 to process operands and operation codes within assembly instructions, aiming to strike a balance between extracting sufficient contextual information and mitigating the OOV issue.

The data normalization preprocessing method proposed in this paper retains sufficient context information of instruction sequences. This method significantly reduces corpus vocabulary size and OOV probability, forming the foundation for inputting high-quality datasets into subsequent models.

### 4.3. BERT-Based Function Embedding (BBFE) Method

According to the characteristics of malicious code assembly code, this paper improves the pre-training process when using the BERT model. According to RoBERTa [41], the Masked Language Model is refined by using dynamic mask technology. With reference to DeepSemantic [20] and OrderMatters [42], the next prediction task is not employed. The specific improvements for the two pre-training tasks are described as follows:

1. The Masked Language Model (MLM) task uses dynamic masks for data enhancement. The original BERT model employs a static masking strategy during the pre-training of the MLM task, wherein a random subset of 15% of the words is initially masked and remains unchanged throughout the entire pre-training process. In contrast, our dynamic masking approach involves re-selecting 15% of the words to be masked at the beginning of each training epoch. Specifically, for each input sequence, a new set of words is randomly chosen for masking before being fed into the model. This dynamic method effectively simulates an expanded training dataset without the need for additional data, thereby mitigating the issue of insufficient training data, which may hinder the model’s full training. As a data augmentation technique, dynamic masking enhances the original data, enabling the model to achieve improved performance and generalization capabilities.

2. Remove the NSP task. The relationship between the functions is determined by the function call, not by their relative positions, which is the main reason for removing the next prediction task. The model may learn a wrong call relationship between functions when the next prediction task is retained, which not only makes the pre-training task meaningless, but may also reduce the performance of the model. Since the relatively rich function call types (standard library function calls, recursive calls, internal calls and external calls, etc.) have been taken into account when standardizing assembly instructions, removing the next prediction task will not cause the model to learn the call information between functions. Due to the removal of NSP tasks, this work has no segment embedding input compared to the original BERT model in the input layer.

To substantiate the efficiency of the augmented function attributes introduced in this manuscript, a manual function feature engineering method has been devised as a baseline. Specifically, function feature extraction is performed on the mentioned assembly language normalization method. And functions are characterized by a set of features, including counts of internal instructions, data transfer instructions, I/O port transfers, destination address transfers, flag transfers, arithmetic operations, logical operations, unconditional jumps, conditional jumps, interrupts, and function calls.

### 4.4. Hierarchical Attention Pooling Graph Neural Network (HAPGNN)

This section provides a detailed introduction of HAPGNN. As shown in Figure 2, the method leverages graph attention mechanisms to update the status of nodes and implements top-k node down-sampling based on a comprehensive scoring of node information during the pooling process. Additionally, the issue of disconnected structures post-pooling is addressed through the incorporation of structural learning.

#### 4.4.1. Graph Attention Layer

Inspired by [43], this paper introduces a graph attention mechanism for the weighted integration of neighborhood information. The importance of node *j* to node *i*, denoted as $e_{ij}$, is calculated as shown in Equation (Equation 1).(1)$$e_{ij}=LeakyReLU\left(a^{T}\left[W{\vec{h}}_{i}∥W{\vec{h}}_{j}\right]\right)$$
where $h_{i}\in R^{f}$ represents the *f*-dimensional feature vector of node *i*, and $W\in R^{f^{\prime }\times f}$ is a shared weight matrix responsible for linearly transforming the features from *f* dimensions to $f^{\prime }$ dimensions. $a\in R^{2f^{\prime }}$ represents the parameter vector of a feedforward neural network, and the symbol || denotes the vector concatenation operation. The activation function LeakyReLU is utilized to introduce nonlinearity. The unnormalized attention coefficients $e_{ij}$ need to be normalized using the softmax function for application during node status updates, and the normalization process is described by Equation (Equation 2).(2)$$\alpha_{ij}={\mathrm{softmax}}_{j}\left(e_{ij}\right)=\frac{\exp(e_{ij})}{\sum_{k\in N_{i}}\exp\left(e_{ik}\right)}$$

The normalized attention coefficients $\alpha_{ij}$ serve as weights for updating node states. By aggregating the information of adjacent nodes with these weights, the hidden representation $h_{i}^{\prime }$ of node *i* is updated. Additionally, to enhance the model’s expressive capability and stabilize the learning process of the self-attention mechanism, this paper introduces a multi-head attention mechanism. This mechanism computes the self-attention scores of nodes using *K* independent attention heads, and then averages the output vectors to improve the model’s generalization ability, as shown in Equation (Equation 3).(3)$$h_{i}^{\prime }=\sigma \left(\frac{1}{K}\sum_{k=1}^{K}\sum_{j\in N_{i}}\alpha_{ij}^{k}W_{k}h_{j}\right)$$

Through the aforementioned mechanism, our model can concentrate on nodes that have a significant impact in the function call graph, thus improving its ability to detect malicious code.

#### 4.4.2. Graph Pooling Layer

The overall pooling process is illustrated in Algorithm 1. To measure the information content of a node in the graph, this paper considers three levels: node, structure, and comprehensive consideration. This article proposes the Node Information Comprehensive Scoring Method to obtain its comprehensive score, Node Importance Value (NIV). And based on NIV, the top-k nodes are selected for down-sampling to obtain nodes containing more information. In addition, the subgraphs formed by the selected nodes may be disconnected from each other, leading to the loss of some structural information, which affects the transmission of information and the update of node embeddings. Therefore, we further address this issue through structural learning methods.
**Algorithm 1** Graph pooling process.**Require:** Aik, Hik, *r***Ensure:** Aik+1, Hik+11:**Node Ranking and Sampling:**2:   Compute Node Importance Values (NIVs):3:     NIVs=σ(degree(Aik))4:     NIVf=σ(MLP(Hik))5:     NIVsf=∥(Iik−Sik)Hik∥16:   Learn Weights and Calculate Final NIV:7:     v=NIVs∥NIVf∥NIVsf8:     ωp=exp(σ(v·zp+bp))∑p=1Pexp(σ(v·zp+bp))9:     NIV=σ(Σp=1Pωp·NIVp)=σ(ws·NIVs+ws·NIVS+ws·NIVS)10:   Select Top-k Nodes:11:     idx=top−rank(NIV,⌈r×nik⌉)12:     Hik+1=Hik(idx,:)13:     Aik+1=Aik(idx,idx)14:**Subgraph Structural Learning:**15:     Eik+1(p,q)=σ(a[Hik+1(p,:)∥Hik+1(q,:)])+λAik+1(p,q),q∈N2−hop(p)16:     Hik+1(p,q)=sparsemax(Eik+1(p,q))=[Eik+1(p,q)−τ(Eik+1(p,:))]+

**Node ranking and sampling:** This work presents an integrative approach to node scoring, encompassing three critical dimensions: the intrinsic attributes of the nodes, the characteristics of the network structure, and the overall performance of the network. Building upon this, this research introduces learnable weight factors and implements a weighted summation methodology to achieve a more nuanced scoring outcome.

**Compute $NIV_{s}$ on structure perspective:** From the perspective of the graph’s topological structure, methods for measuring node importance include node centrality, closeness centrality, betweenness centrality, and eigenvector centrality. Considering the scale, computational cost, and efficiency of the function call graph, this paper uses node centrality $C_{1}$ to construct its importance from a structural perspective. Additionally, following the definition of node degree centrality results in large integer values. To obtain a balanced comprehensive score for nodes, the sigmoid activation function is employed for normalization, ensuring that the value of $NIV_{s}$ ranges from 0 and 1. The calculation method is shown on **Line 3**, where $A_{i}^{k}$ is the adjacency matrix of the i-th graph at the k-th layer, and degree(·) calculates the degree of all nodes.

**Compute $NIV_{f}$ on node perspective:** Besides the topological information of the graph, node features contain more information. Utilizing the node feature vector h obtained in the graph attention layer, the score of nodes in the feature dimension is calculated through a trainable multilayer perceptron, denoted as $NIV_{f}$. The calculation method is shown on **Line 4**, where $H_{i}^{k}$ is the node representation matrix of the k-th layer. MLP(·) consists of one fully connected layer with an output dimension set to 1.

**Compute $NIV_{sf}$ on comprehensive perspective**: Furthermore, we calculate the score considering both node and structural aspects based on node topology information and attribute, using the Hamiltonian distance ${\left|\right|}_{1}$, as $NIV_{sf}=\left|\right|\left(I_{i}^{k}-{\left(D_{i}^{k}\right)}^{-1}A_{i}^{k}\right)H_{i}^{k}{\left|\right|}_{1}$, where $D_{i}^{k}$ represents the diagonal matrix of $A_{i}^{k}$, and $I_{i}^{k}$ represents the identity matrix. Note that with the hierarchical pooling process, the subsequent pooled graph structure $S_{i}^{k}$ satisfies $\sum_{q=1}^{n_{i}^{k}}S_{i}^{k}\left(p,q\right)=1$. Additionally, $D_{i}^{k}=\mathrm{Diag}\left(d_{1},d_{2},\cdots ,dn_{i}^{k}\right)$, and $d_{p}=\sum_{q}^{n_{i}^{k}}S_{i}^{k}\left(p,q\right)$, so $D_{i}^{k}$ degenerates to $I_{i}^{k}$. Therefore, the former equation can finally be simplified, as shown on **Line 5**.

**Learn weights and calculate final NIV**: Due to the potential bias in the ranking of nodes under a single dimension, where the importance of nodes may vary across different dimensions, this paper attempts to aggregate and integrate the rankings of nodes from various dimensions to generate a more robust node ranking. To highlight the differences in node scores from various perspectives, inspired by [44], we adopted a self-attention mechanism-based weight learning approach to autonomously learn the weights derived from each aspect’s score. This step involves a nuanced approach where each dimension’s score $\left(NIV_{s},NIV_{f},NIV_{sf}\right)$ is denoted as $NIV_{p}$ for p=1 to *P*, and the weights are denoted as $\omega_{p}$ for p=1 to *P*. The self-attention mechanism allows the model to dynamically adjust these weights based on the input data, ensuring a more accurate representation of each node’s importance. This is crucial for the subsequent pooling process, as it ensures that the selected nodes for down-sampling are those with the most significant information content and structural roles within the graph.

Finally, the ultimate node importance score NIV is derived based on obtained weights, as shown on **Lines 6–9**, where || means concatenate, and $z_{p}$ and $v_{p}$ are parameter vectors under aspect *p*.

**Select top-k nodes**: After obtaining the NIV of the nodes, the top-k nodes are selected for down-sampling to generate the induced subgraph. The process is shown on **Lines 10–13**, where *r* is the pooling rate of the nodes, and $n_{i}^{k}$ represents the number of nodes in the *i*-th graph at the *k*-th layer. The latter two formulas represent extracting the feature matrix and adjacency matrix of the induced subgraph from the original matrix based on the indices.

At this point, the node features for the next layer have been obtained, while the current structural information $A_{i}^{k+1}$ cannot fully represent the architecture information of the current layer.

**Structural learning mechanism**: The aforementioned pooling operations may cause highly correlated node subgraphs to become disconnected in the induced subgraph, leading to the loss of the graph structure’s integrity. In the context of this paper’s function call graph scenario, an important function node in a sample may become disconnected in the induced subgraph after pooling operations, resulting in the loss of the original call relationships. To address this issue, the work [45] proposed using an approximate distance metric learning algorithm to adaptively estimate the graph Laplacian, which may lead to a locally optimal solution. Research [46] introduced a learned graph structure for node label estimation; however, it generates a dense connected graph, which is not suitable for hierarchical graph representation learning scenarios.

Inspired by [47], this paper proposes a sparse graph structure using a sparse attention mechanism. In addressing graph structural integrity issues, the sparse attention mechanism demonstrates unique advantages over other methods. Firstly, it significantly reduces computational complexity by limiting the number of connections considered within the attention mechanism, particularly in large-scale graph structures, effectively circumventing the computational bottlenecks caused by the $O(n^{2})$ complexity of traditional dense attention mechanisms. For instance, while the Transformer’s full attention mechanism requires calculating attention scores for every node pair, sparse attention mechanisms such as those in the Longformer and Reformer models focus only on a subset of relevant connections, achieving linear complexity and substantially decreasing memory and processing demands. Secondly, the sparse attention mechanism enhances model interpretability. It selectively attends to key connections within the graph, making the model’s decision-making process clearer and more understandable, which is crucial for applications like malware detection that require precise identification of critical graph relationships. The Reformer model utilizes Locality-Sensitive Hashing (LSH) to reduce the amount of attention computation while maintaining performance, facilitating the tracking of the most influential connections. Lastly, this mechanism improves the model’s generalization capabilities. By concentrating on the most relevant information, the model becomes more robust against noise and variations in the graph structure. Compared to dense attention mechanisms that are prone to overfitting irrelevant connections, it can better adapt to dynamically changing graph environments. The goal of this section is to learn a refined graph structure capable of encoding potential pairwise relationships between each pair of nodes.

For the *k*-th layer pooled subgraph $G_{i}^{k}$ of graph $G_{i}$, the proposed learning mechanism uses its structural information $A_{i}^{k}\in R^{n_{i}^{k}\times n_{i}^{k}}$ and hidden representations $H_{i}^{k}\in R^{n_{i}^{k}\times d}$ as inputs. Formally, a single-layer neural network is parameterized by a weight vector $a\in R^{1\times 2d}$. Subsequently, connection score between nodes $v_{p}$ and $v_{q}$ is calculated through an attention mechanism. The process is represented on **Line 15**, where σ(·) is the activation function and || means concatenate. $H_{i}^{k}\left(p,:\right)$ and $H_{i}^{k}\left(q,:\right)$ represent the nodes $v_{p}$ and $v_{q}$, respectively. Specifically, input $A_{i}^{k}$ is the adjacency matrix at the *k*-th layer, where $A_{i}^{k}\left(p,q\right)=0$ indicates that nodes $v_{p}$ and $v_{q}$ are not connected. Introducing $A_{i}^{k}$ ensures that the structural learning mechanism learns potential pairwise relationships between disconnected nodes based on the existing connections. λ can be understood as the balancing parameter.

Inspired by HGP-sl [39], to make connection scores easily comparable among different nodes, the softmax function is used for normalization. But resulting scores still always contain non-zero values, indicating a dense fully connected graph, which does not fit the modeling scenario for adjacency matrices in function call graphs. Therefore, sparsification is needed, as shown on **Line 16**, where τ(·) is used to calculate the sparsification threshold, and ${\left[\cdot \right]}_{+}$ represents the non-negative truncation operation.

Additionally, learning global structural information $S_{i}^{k}$ incurs significant computational costs. In the context of processing function call graphs, exhaustive structural learning for all nodes is unnecessary. Therefore, this paper only conducts structural learning for neighbors within two hops of each node, denoted as $q\in N_{2-hop}\left(p\right)$. Thus, the sparsemax operation is applied only to two-hop neighbors. Consequently, this method yields a refined graph structure $A_{i}^{k+1}$, which is the structural information to be input into the next layer’s stacking module.

#### 4.4.3. Graph Embedding and Classification

**Layer stacking**: Similar to other layer stacking methods in hierarchical graph neural networks, our work stacks multiple layers in the order of graph attention layers followed by graph pooling layers. Specifically, we begin by applying a graph attention layer to the input graph. This layer computes attention weights for each node based on its features and the features of its neighbors, allowing the model to focus on the most relevant information. Following this, a graph pooling layer is applied to aggregate information from the nodes and reduce the graph’s complexity. The pooling layer selects a subset of nodes based on their importance scores, which are learned during training. This process not only reduces the number of nodes but also preserves the most significant structural information. The output from the pooling layer is then passed to the next graph attention layer, and this sequence of attention and pooling layers is repeated multiple times to capture the graph’s hierarchical structure at different levels. This structured stacking of layers enables our model to efficiently learn and represent complex graph structures.

**Readout function and loss function**: The subgraph features of various scales obtained by each pooling layer for graph *i* are $H_{i}^{1},H_{i}^{2},\ldots ,H_{i}^{k}$. To generate a fixed-size graph-level representation, a readout function inspired by [38] is utilized to aggregate all node representations in the subgraphs, resulting in the final embedding $r_{i}$ for graph *i*. This process is shown in Equation (Equation 4).(4)$$\begin{array}{cc} r_{i}^{k} & =\sigma \left(\frac{1}{n_{i}^{k}}\sum_{p=1}^{n_{i}^{n}}H_{i}^{k}\left(p,:\right)∥\max_{q=1}^{d}H_{i}^{k}\left(:,q\right)\right)\\ r_{i} & =r_{i}^{1}+r_{i}^{2}+\cdots +r_{i}^{k} \end{array}$$

Finally, the graph embedding is input into a multilayer perceptron (MLP) for classification, and the loss function is defined as the cross-entropy of the predicted labels, as shown in Equation (Equation 5).(5)$$\begin{array}{cc} \hat{Y} & =\mathrm{softmax}(\mathrm{MLP}(r))\\ L & =-\sum_{i\in G}\sum_{j=1}^{c}Y_{ij}\log{\hat{Y}}_{ij} \end{array}$$
where ${\hat{Y}}_{ij}$ represents the predicted probability that $G_{i}$ belongs to class *j*, and $Y_{ij}$ is the ground truth. *G* denotes the training set of labeled graphs.

## 5. Experiments

### 5.1. Dataset Preparation

#### 5.1.1. Dataset

**Kaggle**: This experimental dataset is from the ‘Microsoft Malware Classification Challenge’ (BIG 2015) on the Kaggle platform, which aims to classify nine malware families. This dataset was chosen for its comprehensive representation of diverse malware families, providing a robust testbed for evaluating classification models. Due to the unavailability of test dataset labels, cross-validation is performed directly on the training dataset in this study. This dataset comprises nine malware families: Ramnit, Lollipop, Kelihos ver3, Vundo, Simda, Tracur, Kelihos_ver1, Obfuscator.ACY, and Gatak. Table 4 illustrates the data distribution of Kaggle.

**VirusShare**: This dataset is from the public data source https://virusshare.com, accessed on 13 January 2024. This dataset was selected for its extensive collection of malware samples, which include a wide variety of malicious software types. This diversity allows us to simulate real-world scenarios by distinguishing between benign and malicious samples effectively. For ease of analysis, all packed samples were filtered out using the PEiD tool, resulting in 3368 unpacked malware samples. Additionally, this study extracted 6809 normal samples from a secure Windows 10 operating system. The specific distribution is shown in Table 5.

#### 5.1.2. Data Preprocessing

The samples in the VirusShare dataset are binary executable files. Firstly, $IDA_{pro}$ is used to disassemble these files to obtain assembly code sequences. The opcodes and operands are then processed according to the standardization method mentioned earlier.

Further extraction of function sequences is required to generate function call graphs. The first step is to identify function boundaries. Specifically, in assembly files, “sub_* proc near” and “sub_* proc far” are considered to be the start of function boundaries, and **endp* marks the end. A dictionary is built to map function names to unique numeric IDs, constructing one-hot coding for functions. The second step is to extract function call relationships and construct function call graphs. Specifically, while processing a single function, when a new function call is detected, the called function’s ID is added to a list. This list records the IDs of all other functions called within a function. Based on the function call lists of all functions, the function call graph of the malware sample can be obtained.

### 5.2. Experiment Setup

#### 5.2.1. Experimental Settings

In the experiment, PyTorch was utilized as the deep learning framework to construct the proposed framework. The runtime environment of experiment is as follows: (1) Intel(R) Core(TM) i7-10870H CPU 393 @2.20 GHz, 16 GB memory, (2) Ubuntu 18.04 (64 bit). To train the model, we set the batch size to 16 and the learning rate of 0.0001.

The computational complexity of the multi-head self-attention mechanism in the BERT model is $O(seq^{2}\cdot d)$, where seq represents the length of the input sequence, and *d* denotes the dimensionality of the vectors. As the input length increases, the computational burden escalates significantly, requiring more substantial computing resources and memory to handle the data. Therefore, to optimize computational efficiency and manage resource utilization, the input length is generally limited. This study analyzes the distribution of function instruction counts using the Kaggle dataset as a basis for investigation. The statistical results are shown in Table 6.

It is observed that 95.6% of function instruction sequence lengths in this dataset are less than 250. Therefore, the length of the input sequence is set to 256.

**Metrics**: Precision, Accuracy, Recall, and F1-score are used to evaluate the proposed method and baseline methods.

**Dataset partitioning**: Dataset is randomly divided into three parts: 60%, 20%, and 20%, corresponding to the training set, validation set, and test set, respectively.

#### 5.2.2. Experimental Design

Overall, to demonstrate the generalization performance of the proposed method, classification experiments on the Kaggle dataset and detection experiments on the VirusShare dataset were conducted. To evaluate the effectiveness of the proposed method as well as the individual components of BBFE and HAPGNN, ablation studies were designed at both the function level and the graph neural network model level.

Additionally, comparisons with other graph neural network models were made, including GCN [14], GAT [43], and GraphSAGE [48]. GCN is a spectral convolution method that aggregates node features with their neighbors’ features through multiplication of the adjacency matrix and node feature matrix. GAT introduces an attention mechanism that allows the model to dynamically focus on more important neighbor nodes for the current task, capturing structural details of the graph more precisely. GraphSAGE generates node embeddings by randomly sampling neighbor nodes and using aggregation functions to combine neighbor features. The reason for focusing on these specific models is that they represent a diverse range of approaches within the graph neural network domain, each with unique strengths and mechanisms for handling graph data. GCN, GAT, and GraphSAGE are widely recognized for their effectiveness in capturing different aspects of graph structures, making them suitable benchmarks for our study.

More importantly, to highlight the effectiveness of the individual components of the proposed graph neural network model, comparisons with other graph pooling models were also made: the global pooling model DGCNN [33] and the hierarchical pooling model SAGPool [29]. DGCNN’s innovation lies in the introduction of the sortPooling mechanism, which defines node order based on topological sorting, allowing traditional one-dimensional convolution to be applied to graph structures to extract features useful for graph classification. SAGPool selects important nodes in the graph for pooling operations through a self-attention mechanism. These models were chosen to represent different pooling strategies and their impact on graph representation, providing a comprehensive evaluation of our proposed pooling mechanism.

Furthermore, this paper also draws comparisons with other state-of-the-art malware detection research. MalConv [49] utilizes a convolutional neural network architecture for malware classification. MAGIC [11] employs deep graph convolutional neural network for malware classification with control flow graph representations. This method focuses on leveraging graph structures to enhance the accuracy of malware classification. MalwareExpert [12] also employs function call graphs, combined with Asm2Vec and SAFE graph embedding techniques, using a graph convolutional network-based model to classify malicious code.

### 5.3. Experiment Results and Analysis

#### 5.3.1. Comparative Analysis of Different Graph Neural Networks

**Approach**: To verify the effectiveness of the proposed models, this paper compares them with other graph neural networks, including different GCNs, Graph Attention Networks, and global and hierarchical pooling graph neural network models. Each of these methods provides a different perspective on how to capture the essence of the graph for our analysis.

The experimental results of different GNN models are shown in Figure 3.

**Results and analysis**: The proposed method outperforms other baseline models in all test datasets. For example, on the Kaggle dataset, our method improves detection accuracy by nearly 3% compared to traditional methods, while on the VirusShare dataset, the accuracy improvement is more significant, reaching nearly 5%.

Compared to GCN, the advantages of other methods lie in their ability to not only capture local neighborhood information but also deepen the understanding of function call relationships through refined sampling and aggregation strategies. This approach enables the model to more effectively extract features from malicious code. For instance, GCN may fail to identify malicious behaviors that propagate through complex call chains, whereas the method presented in this paper can reveal these behaviors through its advanced sampling mechanisms. Although GAT’s attention mechanism provides the model with the ability to focus on important nodes, it may not fully capture all key nodes when dealing with the complex function call graphs of malware. GAT might overlook nodes that play a supporting role in malicious behaviors; while these nodes are not central, they are crucial for understanding the entire attack process. GraphSAGE’s sampling strategy, while innovative, may not fully leverage the structural characteristics of malware in malicious code detection. GraphSAGE may perform poorly when handling malware with highly modular structures, as it may not effectively integrate inter-module function call information.

Compared to DGCNN’s global pooling method, SAGPool, as a hierarchical pooling model, can more effectively handle the hierarchical structure of malicious sample function call graphs. DGCNN may lose critical hierarchical structural information during the pooling process, which can lead to performance degradation in malicious code detection. Through case studies, we have demonstrated how SAGPool improves detection accuracy by preserving the graph’s hierarchical structure.

Building on hierarchical pooling, the method presented in this paper further introduces attention and structure learning modules. The attention module enables the model to dynamically adjust its focus on different nodes, thereby more accurately identifying key functions and potential threats in malicious code detection. The structure learning module allows the model to learn more optimized graph structure representations, which enhances the ability to recognize malicious code and strengthens the model’s generalization capability. These enhancements enable the method in this paper to achieve superior performance in malicious code representation and detection tasks compared to SAGPool.

The above results demonstrate that our method shows significant performance improvement in malware detection tasks, validating the effectiveness and advancement of the proposed method.

#### 5.3.2. Comparative Analysis of Different Function Embedding Strategies

**Approach**: In this section, we delve into the effects of integrating function embeddings into our model. This work proposes that by concurrently analyzing node attributes and the graph’s layout, we can obtain more informative graph embeddings, which should improve the performance of our predictive model. To validate this intuition, we evaluate our approach using various types of function embeddings: the basic manual method and the advanced DeepSemantic technique. In addition, we also examine the results of function embedding across various dimensions.

Particularly, it should be noted that in this experiment, the number of stacked layers in HAPGNN is fixed at k = 3, and the hidden vector dimension is set to 256. Classification experiments are performed on the Kaggle dataset, and detection experiments are conducted on VirusShare.

**Results and analysis:** The experimental results of different function embedding methods and dimensions are shown in Table 7.

Observing the detection performance of different function embedding methods, results show that the BBFE-based method is significantly better than the manual method, indicating that the proposed method can extract richer function-level semantic features, which directly improves the detection results. A small dimension of the embedding vector will result in weak model representation capability, meaning the model cannot effectively capture the semantic information in the input sequence. As the dimension increases, there is a corresponding increase in the number of model parameters and computational complexity. However, the performance metrics, as shown in Table 7, indicate that the optimal embedding dimension is 128, where the model achieves a balance between accuracy and computational efficiency. Dimensions beyond 128 do not offer significant performance improvements, suggesting that the increase in model parameters and complexity does not contribute to enhanced generalization capability.

#### 5.3.3. Comparative Analysis of Preprocessing Effects

**Approach**: It is posited in this paper that the granularity of instruction standardization determines the richness of information contained in the model’s input data, which in turn affects the performance of the dataset. Therefore, the focus of this part is on the impact of instruction standardization methods on the experiment. For ease of analysis, the dataset from Kaggle was standardized in terms of opcodes and operands, and OOV occurrences were tallied. The results are presented in Table 8 below.

**Results and analysis:** Initially, the total vocabulary size reached an extremely large figure of 1.58 million before any standardization was applied. Upon the standardization of operands, the vocabulary size was dramatically curtailed to around 700,000. Further, subsequent to the standardization of operands, the vocabulary size experienced a considerable reduction, settling at a mere 5000. The standardization of opcodes and operands significantly reduced the OOV ratio from 4.18% to 0.08%, highlighting the strength of our preprocessing in managing OOV occurrences. The normalization process achieved superior results, with 98.90% for Kaggle and 99.57% for VirusShare, outperforming non-standardized and partially standardized methods. These findings underscore the critical role of comprehensive data preprocessing in enhancing model performance by effectively addressing OOV issues.

#### 5.3.4. Comparative Analysis of Different Model Settings

**Approach**: This section explores how key settings in HAPGNN influence the model’s performance in malware detection tasks. Optimizing settings is a critical step in enhancing model performance, preventing overfitting, and improving generalization capability. To verify that our proposed approach does not have over-squashing and over-smoothing issues, we conduct performance evaluations using different stacking layers and observe whether the performance is stable. Additionally, the dimensions of the hidden layer were adjusted, starting from smaller dimensions and gradually increasing to observe changes in the model’s learning capability. Based on the above experiments, this paper chose to continue with 128-dimensional function features based on BBFE. Specifically, the model was trained and evaluated with different hidden unit sizes of 64, 128, 256, and 512. Comparative experiments were also conducted with the number of stacked layers set to 1, 3, and 5.

**Results and analysis:** The experimental results of different HAPGNN settings are shown in Table 9.

The enhancement in model performance as the number of hidden units increases from 64 to 256 can be attributed to the model’s improved capacity to capture intricate feature interactions. With the expansion in dimensionality, the model’s expressive power is augmented, enabling it to discern subtler patterns within the data. However, a decline in performance is observed when the number of hidden units is further increased to 512, potentially due to the model’s increased complexity leading to overfitting. In such cases, the model may learn the noise in the training data rather than the underlying data distribution, thereby compromising its generalization capabilities.

Introducing additional layers can elevate the model’s expressive capabilities, facilitating the capture of deeper-level features and complex patterns. Nonetheless, an excessive number of layers may precipitate issues such as vanishing or exploding gradients, which can adversely affect the model’s training efficiency and generalization. The methodology presented in this paper carefully calibrates the number of stacked layers to circumvent these challenges, thereby maintaining model performance while enhancing training efficiency and generalization.

#### 5.3.5. Comparison with Other Methods

**Approach**: To demonstrate the effectiveness of our proposed method, this paper compares it with several advanced state-of-the-art works, specifically comparing different feature engineering methods, detection models, datasets, and accuracy results, and delves into the deeper reasons.

**Results and analysis**: The comparisons of different methods are shown in Table 10.

MalConv [49] takes the bytes of raw files as input, extracts features using embedding layers and convolutional layers, and classifies using fully connected layers. Its accuracy is relatively low due to the lack of deep exploration of the intrinsic features of malicious code.

MAGIC [11] extended the existing DGCNN model and introduced their own improvements. Our method proposed in this paper is based on function call graphs, which consumes fewer data preprocessing resources than MAGIC’s CFG-based approach while achieving higher accuracy.

MalwareExpert [12] achieved 97.3% accuracy and 96.5% recall on the unpacked VirusShare dataset. However, it does not further normalize the data during function node embedding, and the GCN treats all nodes equally, leading to slower learning efficiency.

In summary, this paper achieves graph representation by employing graph attention and hierarchical pooling mechanisms after pre-embedding functions. Initially, BBFE is used to pre-embed functions, retaining the rich semantic information of the functions themselves. Simultaneously, the HAPGNN model is capable of extracting more information, demonstrating excellent results across various datasets.

## 6. Conclusions and Future Work

This study proposes a hierarchical attention pooling graph neural network method for malware detection, enhancing the effectiveness of malware detection by extracting deep semantic and structural information from enhanced function call graphs. Experimental results indicate that the proposed method significantly outperforms other methods on multiple datasets, including Kaggle and VirusShare. Its superior performance is primarily attributed to the following key aspects:

1. High semantic preservation function embedding: This paper presents a BBFE method that effectively tackles the OOV issue, ensuring the preservation of semantic richness within function call graphs. This innovation is pivotal to our framework’s ability to detect malware with heightened accuracy and robustness.

2. Effective graph neural networks: We introduce an advanced graph pooling mechanism that achieves more rational node ranking through a comprehensive node information scoring method. Additionally, by incorporating a structural learning mechanism, this approach ensures the connectivity of the subgraphs after pooling, preserving key structural information.

In summary, our research demonstrates the potential of graph neural networks in improving malware detection technology. By combining advanced graph learning with static analysis, this work offer a powerful tool for identifying malware threats.

Future work will focus on refining our proposed method to provide even more detailed insights into malware functions, thereby enhancing the interpretability of malware detection results. This will ultimately support the key objective of automatically generating comprehensive analysis reports.

## Figures and Tables

**Figure 1 sensors-25-00374-f001:**
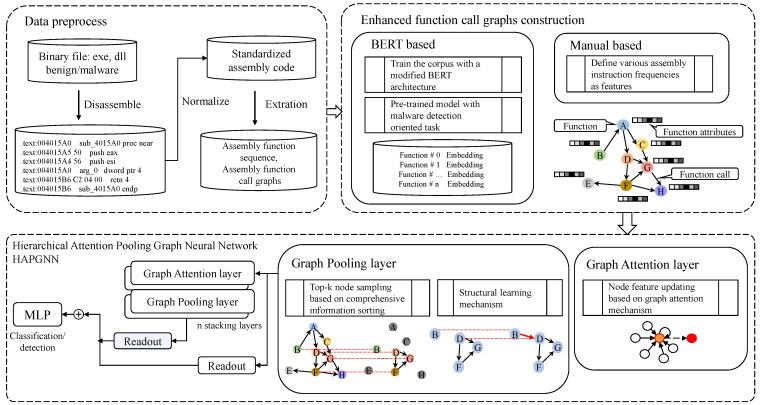
Overall architecture of MalHAPGNN.

**Figure 2 sensors-25-00374-f002:**
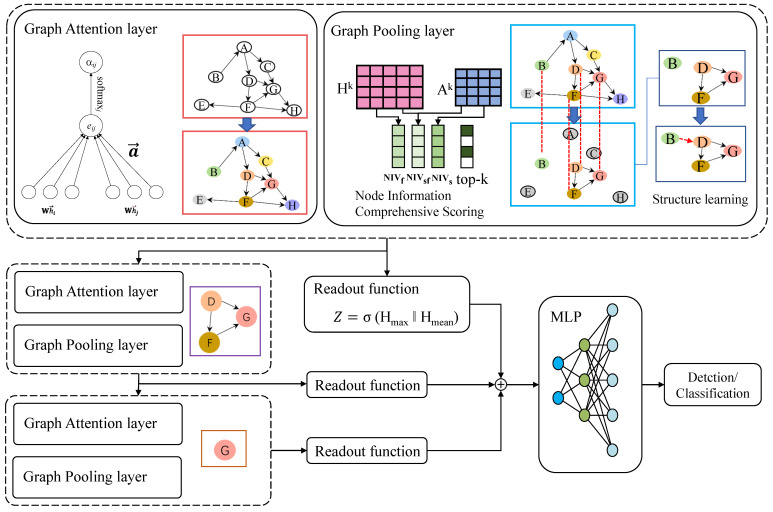
Overview of HAPGNN.

**Figure 3 sensors-25-00374-f003:**
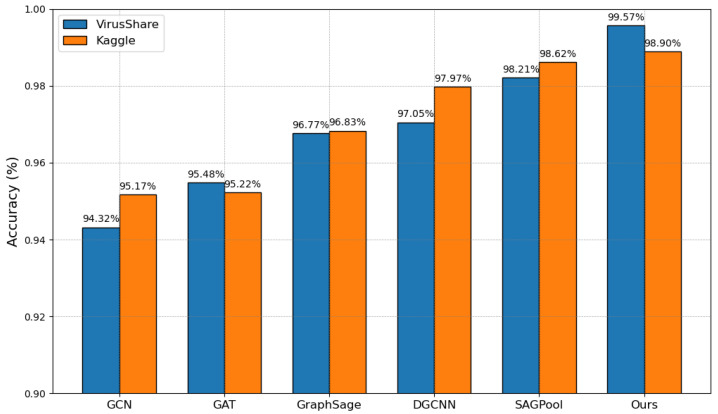
Performance comparison of different GNN models.

**Table 1 sensors-25-00374-t001:** Comparison of different graph pooling methods.

Methods	Pooling Description	Limitations
DiffPool [27]	Soft clustering of nodes	High computation cost
gPool [28]	Selects top-k nodes	Does not utilize the graph structure
SAGPool [29]	Uses self-attention mechanism to decide node retention	Considers node features and graph structure, but high computation cost
EdgePool [30]	Pools edges in the graph	Flexible, but usually pools about half the nodes
DGCNN [33]	Extracts local features through convolution and pool according to feature map values	Global pooling, treating all nodes equally

**Table 2 sensors-25-00374-t002:** Normalization rules for instruction opcodes.

Normalized Opcode	Opcode Contained
DataTransferAssignment	mov, movx, moxzx, xadd
DataTransferExchange	bswap, xchg, cmpxchg, xlat
DataTransferStack	push, pop, pusha, popa, pushad, pushas
I/OPortTransfer	in, out
AddressTransfer	lea, lds, les, lfs, lgs, lss
FlagTransferAH	lahf, sahf
FlagTransferStack	pushf, popf, pusd, popd
GeneralArithmetic	add, adc, inc, sub, sbb, dec
	mul, imul, div, idiv
Compare	cmp
GeneralLogicalOperation	and, or, xor, not

**Table 3 sensors-25-00374-t003:** Normalization rules for instruction operands.

Oprand	Description	Normalized Oprand
Immediate	standard library function	libc_func
	recursive call	self_func
	inner function	inner_func
	external function	extern_func
	jump address	jmp_dst
	string	disp_str
	statically allocated variables	disp_bss
	data other than string	immval
Register	size	reg[*]
	stack/base/instruction pointer	[s|b|i]p[*]
	flags	reg[c|r|d|st], reg[c|d|e|f|s]
	advanced vector extensions	reg[x|y|z]*mm
	direct	memptr[*]
Pointer	direct addressing	memptr[*]
	indirect addressing	[base+index*scale+disp]

**Table 4 sensors-25-00374-t004:** Distribution of samples in Kaggle.

Category	Quantity
Ramnit	1541
Lollipop	2478
Kelihos_ver3	2942
Vundo	475
Simda	42
Tracur	751
Kelihos_ver1	398
Obfuscator.ACY	1228
Gatak	1013
Total	10,868

**Table 5 sensors-25-00374-t005:** Distribution of samples in VirusShare.

Category	Quantity
Malicious	3368
Benign	6809
Total	10,177

**Table 6 sensors-25-00374-t006:** Distribution of the quantity of instructions per function.

Function Instruction Quantity Range	Function Quantity	Proportion	Accumulated
0–10	528,855	26.57%	26.57%
10–50	881,279	44.28%	70.85%
50–100	280,610	14.10%	84.95%
100–150	115,321	5.79%	90.75%
150–200	55,510	2.79%	93.53%
200–250	31,671	1.59%	95.13%
250+	96,999	4.87%	100.00%

**Table 7 sensors-25-00374-t007:** Performance comparison of different function embedding approaches.

Dataset	Function Embedding	Dimension	Accuracy	Precision	Recall	F1-Score
Kaggle	Manual	15	95.83%	97.12%	96.98%	97.05%
	DeepSemantic	128	96.81%	98.13%	97.99%	98.32%
		64	97.86%	97.98%	97.79%	97.88%
	BBFE (ours)	128	**98.90%**	**98.96%**	**98.78%**	**98.87%**
		256	98.86%	98.32%	98.13%	98.22%
VirusShare	Manual	15	97.18%	97.65%	97.92%	97.78%
	DeepSemantic	128	99.02%	98.75%	98.23%	98.55%
		64	98.61%	98.01%	98.39%	98.20%
	BBFE (ours)	128	**99.57%**	**98.53%**	**99.15%**	**98.84%**
		256	98.94%	98.26%	98.70%	98.48%

**Table 8 sensors-25-00374-t008:** Summary of normalized results under different granularities.

Normalized Object	Vocabulary Quantity			OOV Quantity		Kaggle	VirusShare
	Total	Train	Test	Amount	Proportion		
none	1,588,377	1,571,594	824,127	34,480	4.18%	95.61%	93.72%
opcode only	889,313	877,949	438,479	2244	0.51%	96.77%	95.32%
operand only	729,696	719,995	352,461	2323	0.66%	96.89%	96.41%
opcode and operand	5481	5476	4784	4	0.08%	**98.90%**	**99.57%**

**Table 9 sensors-25-00374-t009:** Performance comparison of different HAPGNN settings on VirusShare and Kaggle datasets.

HAPGNN Stacking Layers	HAPGNN Hidden Dimension	VirusShare				Kaggle			
		Accuracy	Precision	Recall	F1-Score	Accuracy	Precision	Recall	F1-Score
1	64	95.37%	96.01%	95.99%	96.50%	97.61%	97.77%	97.85%	97.81%
	128	96.68%	96.34%	96.51%	96.92%	97.79%	97.94%	97.98%	97.96%
	256	96.28%	96.40%	96.92%	96.46%	**98.16%**	**98.38%**	**98.35%**	**98.37%**
	512	**97.16%**	**97.23%**	**97.60%**	**97.41%**	97.96%	98.12%	98.17%	98.14%
3	64	97.87%	97.81%	97.97%	97.89%	97.75%	97.96%	97.67%	97.81%
	128	98.32%	98.24%	98.15%	98.19%	97.86%	97.98%	97.89%	97.93%
	256	**99.01%**	**98.41%**	**98.78%**	**98.59%**	**98.45%**	**98.62%**	**98.60%**	**98.61%**
	512	98.98%	98.28%	98.41%	98.34%	98.07%	98.28%	98.23%	98.25%
5	64	98.13%	98.21%	97.92%	98.06%	97.92%	97.91%	97.88%	97.89%
	128	98.43%	98.28%	98.39%	98.33%	98.15%	98.21%	98.03%	98.12%
	256	**99.57%**	**98.53%**	**99.15%**	**98.84%**	**98.90%**	**98.96%**	**98.78%**	**98.87%**
	512	98.85%	98.45%	98.92%	98.68%	98.72%	98.64%	98.39%	98.51%

**Table 10 sensors-25-00374-t010:** Performance comparison with other methods.

Methods	Feature	Model	Accuracy
MalConv [49]	Raw byte sequence	CNN	89.6% (VirusShare)
MAGIC [11]	CFG and statistical features	GNN	98.25% (Kaggle)
MalwareExpert [12]	Function call graph	GNN, Asm2Vec, SAFE	97.3% (VirusShare)
MalHAPGNN	Function call graph	MalHAPGAN	**99.57%** (VirusShare) | **98.90%** (Kaggle)

## Data Availability

The data used to validate the outcomes of this investigation can be obtained by contacting the corresponding author.

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
