# Peer review of "MalHAPGNN: An Enhanced Call Graph-Based Malware Detection Framework Using Hierarchical Attention Pooling Graph Neural Network"

_sensors, 2025, doi:10.3390/s25020374_

Round 1
Reviewer 1 Report
Comments and Suggestions for Authors
1,The differences between MVPool and the graph pooling operation proposed in this article should be explained in detail, emphasizing the innovation of this article.
2,In the process of data preprocessing, please provide additional information on what operations are included in "comprehensive data cleaning" and how these operations affect subsequent analysis.
3,In the graph pooling layer, it is necessary to provide a detailed explanation of how the weight factors of different node importance calculation methods (such as based on node centrality, feature dimension scores, etc.) are learned and adjusted, and provide algorithm details.
4,In terms of structural learning mechanism, please compare and analyze the advantages of the sparse attention mechanism used in this article compared to other similar methods in solving graph structural integrity problems.
5,In the preparation of the dataset, explain the specific reasons for choosing the Kaggle and VirusShare datasets, as well as their representativeness in terms of malware types, sample characteristics, and other aspects.
6,Based on the experimental results of different GNN models, it is necessary to deeply analyze the specific reasons for the poor performance of other models, combined with the model structure and characteristics of malicious software data, and illustrate through case studies.
7,The conclusion can further summarize the key findings and lessons learned during the research process, providing more valuable references for subsequent research.
Author Response
Thank you very much for taking the time to review this manuscript. Please find the detailed responses below and the corresponding revisions/corrections highlighted/in track changes in the re-submitted files.
We sincerely thank your positive and valuable comments on our manuscript.
1,The differences between MVPool and the graph pooling operation proposed in this article should be explained in detail, emphasizing the innovation of this article.
Thank you for your constructive comment. We have elaborated on the distinctions between MVPool and our proposed graph pooling operation, highlighting the novel aspects of our approach.
There are two primary differences: firstly, the state update mechanism within the same layer, which refers to the internal processing of information within a single layer, and secondly, the inter-layer information transfer mechanism, which concerns the way information is conveyed across different layers.
On one hand, regarding intra-layer state updates, we have incorporated a graph attention mechanism. This is particularly applicable in the context of malware detection, where it is crucial that functions of greater significance in the function call graph receive increased focus.
On the other hand, concerning inter-layer information transfer, a pivotal component is node ranking. We have designed a Node Importance Value (NIV) calculation method tailored for function nodes in malware detection scenarios. This has enabled us to implement efficient sampling.
We have included the aforementioned information in the final paragraph of the related work section.
"Based on the aforementioned analysis, we propose that the pooling component can be enhanced in two specific respects. First, concerning the state update within layers, graph attention algorithms may be particularly well-suited to scenarios in malicious code detection where functions of greater significance within the call graph require heightened attention. Second, with respect to the propagation of information between layers, there is a necessity for an improved method of calculating Node Importance Value for function nodes in the context of malicious code detection, which would facilitate more efficient sampling."
2,In the process of data preprocessing, please provide additional information on what operations are included in "comprehensive data cleaning" and how these operations affect subsequent analysis.
Thank you for your valuable comments. We have addressed the concerns regarding data preprocessing and its impact on subsequent analysis as follows:
We have expanded the description of the data preprocessing steps and added a new section to analyze the effects of these operations on the model's performance. This ensures that the readers can better understand the significance of each preprocessing step and its contribution to the overall results.
In Section 5.1.2, we have provided a detailed description of the data preprocessing steps: "The first step is to identify function boundaries. Specifically, in assembly files, 'sub_* proc near' and 'sub_* proc far' are considered to be the start of function boundaries, and '*endp' marks the end. A dictionary is built to map function names to unique numeric IDs, constructing one-hot coding for functions."
We have added a new subsection, "5.3.3. Comparative Analysis of Preprocessing Effects," which includes the following content: "Approach: It is posited in this paper that the granularity of instruction standardization determines the richness of information contained in the model's input data, which in turn affects the performance of the dataset. Therefore, the focus of this part is on the impact of instruction standardization methods on the experiment. For ease of analysis, the dataset from Kaggle has been standardized in terms of opcodes and operands, and OOV occurrences have been tallied." The results are presented in the Table in our paper.
We also have included a deeper analysis in Section 5.3.3: "Initially, the total vocabulary size reached an extremely large figure of 1.58 million before any standardization was applied. Upon the standardization of operands, the vocabulary size was dramatically curtailed to around 700,000. Further, subsequent to the standardization of operands, the vocabulary size experienced a considerable reduction, settling at a mere 5,000. The standardization of opcodes and operands significantly reduces the OOV ratio from 4.18% to 0.08%, highlighting the strength of our preprocessing in managing OOV occurrences. The normalization process achieves superior results, with 98.90% for Kaggle and 99.57% for VirusShare, outperforming non-standardized and partially standardized methods. These findings underscore the critical role of comprehensive data preprocessing in enhancing model performance by effectively addressing OOV issues."
In addition, the analysis of the BERT model's input parameters has been relocated and expanded in Section 5.2.1, "Experimental Settings," with a more accurate description provided. The revised text is as follows:
"The computational complexity of the multi-head self-attention mechanism in the BERT model is O(seq^2 \cdot d), where seq denotes the length of the input sequence and d represents the dimensionality of the vectors. As the input length increases, the computational load grows rapidly, necessitating additional computing resources and memory to process the data. Consequently, to balance computational efficiency and resource consumption, the input length is typically constrained. This paper examines the distribution of function instruction counts using the Kaggle dataset as a reference."
These revisions provide a clearer understanding of how data preprocessing affects subsequent analysis and model performance.
3,In the graph pooling layer, it is necessary to provide a detailed explanation of how the weight factors of different node importance calculation methods (such as based on node centrality, feature dimension scores, etc.) are learned and adjusted, and provide algorithm details.
Thank you for pointing out this issue. We have expanded the explanation in the graph pooling layer to provide a comprehensive understanding of how the weight factors for different node importance calculation methods are learned and adjusted. Here is the revised content:
"Due to the potential bias in the ranking of nodes under a single dimension, where the importance of nodes may vary across different dimensions, this paper attempts to aggregate and integrate the rankings of nodes from various dimensions to generate a more robust node ranking. To highlight the differences in node scores from various perspectives, inspired by \cite{bahdanau_neural_2016}, we adopted a self-attention mechanism-based weight learning approach to autonomously learn the weights derived from each aspect's score.
This step involves a nuanced approach where each dimension's score (NIVs, NIVf, NIVsf) is denoted as {NIVp} for p=1 to P, and the weights are denoted as {ω_p} for p=1 to P. The self-attention mechanism allows the model to dynamically adjust these weights based on the input data, ensuring a more accurate representation of each node's importance. This is crucial for the subsequent pooling process, as it ensures that the selected nodes for downsampling are those with the most significant information content and structural roles within the graph."
4,In terms of structural learning mechanism, please compare and analyze the advantages of the sparse attention mechanism used in this article compared to other similar methods in solving graph structural integrity problems.
Thank you for your insightful comment.
We have expanded our discussion on the structural learning mechanism to include a comparative analysis of the sparse attention mechanism against other similar methods in addressing graph structural integrity issues. Here is the revised content in page 11:
"In addressing graph structural integrity issues, the sparse attention mechanism demonstrates unique advantages over other methods. Firstly, it significantly reduces computational complexity by limiting the number of connections considered within the attention mechanism, particularly in large-scale graph structures, effectively circumventing the computational bottlenecks caused by the $O(n^2)$ complexity of traditional dense attention mechanisms. For instance, while the Transformer's full attention mechanism requires calculating attention scores for every node pair, sparse attention mechanisms such as those in the Longformer and Reformer models focus only on a subset of relevant connections, achieving linear complexity and substantially decreasing memory and processing demands.
Secondly, the sparse attention mechanism enhances model interpretability. It selectively attends to key connections within the graph, making the model's decision-making process clearer and more understandable, which is crucial for applications like malware detection that require precise identification of critical graph relationships. The Reformer model utilizes Locality-Sensitive Hashing (LSH) to reduce the amount of attention computation while maintaining performance, facilitating the tracking of the most influential connections.
Lastly, this mechanism improves the model's generalization capabilities. By concentrating on the most relevant information, the model becomes more robust against noise and variations in the graph structure. Compared to dense attention mechanisms that are prone to overfitting irrelevant connections, it can better adapt to dynamically changing graph environments."
The goal of this section is to learn a refined graph structure capable of encoding potential pairwise relationships between each pair of nodes.
5,In the preparation of the dataset, explain the specific reasons for choosing the Kaggle and VirusShare datasets, as well as their representativeness in terms of malware types, sample characteristics, and other aspects.
Thank you for your comment. We have updated the dataset preparation section to include specific reasons for selecting the Kaggle and VirusShare datasets, along with their representativeness in various aspects. We have added the following details:
"The Kaggle dataset was chosen for its comprehensive representation of diverse malware families, which have undergone thorough disassembly processes, including complete unshelling. This dataset includes a variety of families across nine different types, providing a robust testbed for evaluating classification models."
"The VirusShare dataset was selected for its extensive collection of malware samples, which encompass a broad spectrum of malicious software types. This dataset is more reflective of real-world scenarios, as it includes a higher diversity of samples that are commonly encountered in the wild. This diversity allows us to effectively distinguish between benign and malicious samples, thereby enhancing the model's ability to generalize to new, unseen threats."
6,Based on the experimental results of different GNN models, it is necessary to deeply analyze the specific reasons for the poor performance of other models, combined with the model structure and characteristics of malicious software data, and illustrate through case studies.
Thank you for your constructive comment.
We have expanded Section 5.3.1 "Evaluation of Effectiveness" to include a detailed analysis of the reasons for the suboptimal performance of other models. This analysis delves into the structural and semantic characteristics of malware data and how they affect the effectiveness of various GNN models. We have also included case studies to illustrate these points. Specifically, we added:
"Compared to GCN, the advantages of other methods lie in their ability to not only capture local neighborhood information but also deepen the understanding of function call relationships through refined sampling and aggregation strategies. This approach enables the model to more effectively extract features from malicious code. For instance, GCN may fail to identify malicious behaviors that propagate through complex call chains, whereas the method presented in this paper can reveal these behaviors through its advanced sampling mechanisms. Although GAT's attention mechanism provides the model with the ability to focus on important nodes, it may not fully capture all key nodes when dealing with the complex function call graphs of malware. GAT might overlook nodes that play a supporting role in malicious behaviors; while these nodes are not central, they are crucial for understanding the entire attack process. GraphSAGE's sampling strategy, while innovative, may not fully leverage the structural characteristics of malware in malicious code detection. GraphSAGE may perform poorly when handling malware with highly modular structures, as it may not effectively integrate inter-module function call information.
Compared to DGCNN's global pooling method, SAGPool, as a hierarchical pooling model, can more effectively handle the hierarchical structure of malicious sample function call graphs. DGCNN may lose critical hierarchical structural information during the pooling process, which can lead to performance degradation in malicious code detection. Through case studies, we have demonstrated how SAGPool improves detection accuracy by preserving the graph's hierarchical structure.
Building on hierarchical pooling, the method presented in this paper further introduces attention and structure learning modules. The attention module enables the model to dynamically adjust its focus on different nodes, thereby more accurately identifying key functions and potential threats in malicious code detection. The structure learning module allows the model to learn more optimized graph structure representations, which enhances the ability to recognize malicious code and strengthens the model's generalization capability. These enhancements enable the method in this paper to achieve superior performance in malicious code representation and detection tasks compared to SAGPool.”
7,The conclusion can further summarize the key findings and lessons learned during the research process, providing more valuable references for subsequent research.
Thank you for your insightful comment.
We have expanded the conclusion to further summarize the key findings and lessons learned during our research process. This includes the importance of semantic preservation in function embeddings and the effectiveness of our hierarchical graph pooling mechanism.
Specifically, we have added a summary of the high semantic preservation function embedding:
"High semantic preservation function embedding: This paper presents the BBFE method that effectively tackles the OOV issue, ensuring the preservation of semantic richness within function call graphs. This innovation is pivotal to our framework's ability to detect malware with heightened accuracy and robustness."
Additionally, we have added the following to our future work section:
"Future work will focus on refining our proposed method to provide even more detailed insights into malware functions, thereby enhancing the interpretability of malware detection results. This will ultimately support the key objective of automatically generating comprehensive analysis reports."
We believe these additions will provide valuable references for subsequent research.

Reviewer 2 Report
Comments and Suggestions for Authors
In this paper, authors innovatively introduce a hierarchical attention pooling graph neural network method that enhances call graphs with a BERT-based attribute-enhanced function embedding method to improve semantic features. They also utilize attention mechanisms and pooling operations to extract structural information in order to enhance the effectiveness of malware detection.
Some detailed comments:
1. Abbreviations should be standardized. For abbreviated nouns that appear for the first time, the full name should be added, such as BERT (Bidirectional Encoder Representations from Transformers) and GNN (Graph Neural Network), as seen in lines 7 and 13. Also, the full name of BERT in line 65 and the full name of APT in line 130.
2. In lines 502 - 503, abbreviations of the terms “Graph Convolutional Networks” should be used.
3. Please maintain consistency in the use of name abbreviations.
4. In line 208, does Hi belong to Gi? Does its font need to be bolded?
5. The sentences in lines 318 - 322 are repetitive. It is recommended to make revisions.
6. The subscript of "NIVsf" in line 359 is inconsistent with that in Algorithm 1.
7. "Line 6 - 8" in line 371 should be "Line 6 - 9". The same problem occurs in line 374.
8. In line 506, the word "different" is repeated.
9. In the comparison results, avoid using only "Ours" to refer to your method. Instead, add the name of the proposed network.
10. The data in Table 6 and Figure 3 are Completely repetitive. It is recommended to retain only Figure 3.
11. There are incomplete expressions in the last paragraph. Please correct them, as seen in line 652.
12. In Table 9, is it "oprand" or "operand"?
13. The font in Table 9 is not consistent with that of the other tables and needs to be adjusted to a uniform font size.
14. Kaggle is a platform that contains not only data but also models and code. Therefore, the description of the data source on the Kaggle platform should be made more accurate. For example, use the specific dataset name instead of just "Kaggle".
15. In line 93, the out - of - vocabulary (OOV) issue should be fully explained before, as well as how it affects malware detection.
16. In line 170, how does hierarchical pooling solve the problem? Please elaborate instead of just briefly mentioning it.
17. What does "lstm" in line 176 refer to? Please explain.
18. Can the text in lines 200 - 206 be supplemented with an image? An image would be clearer and more straightforward.
19. In the 5.2.1. Experimental environment, please state whether the environmental configurations on different machines are consistent.
20. In the 5.2.2. Experimental design, please explain the reason for only comparing some of the studies presented in Related work.
21. The subtitle "4.2. Enhanced call graph construction" in line 238 seems a bit ambiguous. Because the content under this subtitle includes not only data preprocess but also BERT - based function call graphs construction. You should describe each part separately according to Figure 1 instead of mixing them together. The same phenomenon occurs in "4.2.1. Assembly function normalization".
22. Lines 272 - 281 mention using dynamic mask to improve the BERT model for data augmentation. This seems to be a difference between your method and other methods. You should describe the steps of dynamic mask instead of only stating its advantages. The same problem exists in other steps and should be comprehensively introduced. For example, the layers stacking in lines 418 - 420.
23. It is recommended to revise the analysis in "5.3.1. Evaluation of Effectiveness". The comparison between different methods should be more explicit, and more attention should be paid to "the essence of the graph" mentioned in lines 504 - 505. Also, in lines 516 - 520 there seems to be some analysis that is incoherent.
24. In "5.3.2. Impacts of Function Embeddings", why not compare with all the feature embedding methods in Related work?
25. The content in lines 538 - 561 seems to be analysis rather than the experimental method that should be written.
26. The description in lines 573 - 582 does not correspond to the content of Table 10. For example, the model parameters and computational complexity increase with the increase of dimension.
27. Lines 598 - 610 only describe the results but lack analysis.
28. It is recommended to add a comparative experiment to analyze and compare the BERT method with other feature embedding methods.
29. Overall, it is recommended that the author reorganize and revise the entire content of "5.3. Experiment results and analysis".
30. In the first point of 6. Conclusion and Future work, it seems that data preprocessing makes the main contribution rather than the function call graph construction method. If the function call graph construction method does not include data preprocessing, then the contribution of data preprocessing should be described separately.
31. The English proficiency of the manuscript needs to be improved.
Author Response
Thank you very much for taking the time to review this manuscript. Please find the detailed responses below and the corresponding revisions/corrections highlighted/in track changes in the re-submitted files.
We sincerely thank your positive and valuable comments on our manuscript.
- Abbreviations should be standardized. For abbreviated nouns that appear for the first time, the full name should be added, such as BERT (Bidirectional Encoder Representations from Transformers) and GNN (Graph Neural Network), as seen in lines 7 and 13. Also, the full name of BERT in line 65 and the full name of APT in line 130.
Thank you for your careful review,here are the specific changes we have made:
We have ensured that the full name "Bidirectional Encoder Representations from Transformers" is provided alongside the abbreviation "BERT" when it first appears in the text. This has been corrected in lines 7.
And we have added the full name "Graph Neural Network" next to the abbreviation "GNN" on its first occurrence in line 13.
And we have included the full name "Advanced Persistent Threat" next to the abbreviation "APT" when it first appears in line 130.
In addition, we standardized for subsequent non-first abbreviations.
- In lines 502 - 503, abbreviations of the terms “Graph Convolutional Networks” should be used.
Thank you for your comment. We have made the necessary adjustment to use the abbreviation "GCNs" for "Graph Convolutional Networks" in lines 502 and 503 as suggested.
- Please maintain consistency in the use of name abbreviations.
Thank you for your observation. We have reviewed the entire manuscript and ensured consistency in the use of abbreviations for names throughout the document.
- In line 208, does Hi belong to Gi? Does its font need to be bolded?
Thank you for your query. Yes, Hi indeed belongs to Gi. However, upon careful consideration, we have determined that Hi does not need to be bolded in line 208.
We have unbolded the original bold Hi.
The notation remains clear and consistent without bolding, and it maintains the readability of the mathematical expressions in the text.
- The sentences in lines 318 - 322 are repetitive. It is recommended to make revisions.
Thank you for pointing out the redundancy in the manuscript.
The revised passage now provides the necessary information more efficiently.
We modify "Furthermore, to enhance the expressive capability of the model and stabilize the learning process of the self-attention mechanism, this paper introduces a multi-head attention mechanism.
Additionally, to enhance the model's expressive capability and stabilize the learning process of the self-attention mechanism, this paper introduces a multi-head attention mechanism." to "Additionally, to enhance the model's expressive capability and stabilize the learning process of the self-attention mechanism, this paper introduces a multi-head attention mechanism.".
- The subscript of "NIVsf" in line 359 is inconsistent with that in Algorithm 1.
Thank you for noticing the inconsistency. We have corrected the subscript of "NIVsf" in line 359 to match the notation used in Algorithm 1.
- "Line 6 - 8" in line 371 should be "Line 6 - 9". The same problem occurs in line 374.
Thank you for identifying the error. The references to line numbers have been corrected from "Line 6 - 8" to "Line 6 - 9" in line 371, and the same correction has been applied in line 374.
- In line 506, the word "different" is repeated.
Thank you for your attention to detail. The repetition of the word "different" in line 506 has been addressed by removing the redundant instance.
- In the comparison results, avoid using only "Ours" to refer to your method. Instead, add the name of the proposed network.
Thank you for your suggestion. We have revised the comparison results to refer to our method by its full name, "MalHAPGNN," in addition to "Ours," to provide clarity and specificity.
- The data in Table 6 and Figure 3 are Completely repetitive. It is recommended to retain only Figure 3.
We appreciate your feedback. In response, we have removed the redundant data from Table 6 and retained only Figure 3 to avoid repetition and enhance the presentation of our results.
- There are incomplete expressions in the last paragraph. Please correct them, as seen in line 652.
Thank you for drawing our attention to the incomplete expressions in the last paragraph. We have corrected the sentence for clarity and coherence. The revised sentence now reads:
"Automatically generating analysis reports for malware is a long-term goal for all malware analysts. Our proposed method could further analyze subgraphs and nodes after pooling and downsampling to support the interpretability of malware detection results."
- In Table 9, is it "oprand" or "operand"?
Thank you for your comment. Upon review, we have identified that the correct term is "operand" and not "oprand." This error has been corrected in Table 9.
- The font in Table 9 is not consistent with that of the other tables and needs to be adjusted to a uniform font size.
Thank you for your observation. The font inconsistency in Table 9 has been corrected, and it is now adjusted to match the font size and style of the other tables in the manuscript.
- Kaggle is a platform that contains not only data but also models and code. Therefore, the description of the data source on the Kaggle platform should be made more accurate. For example, use the specific dataset name instead of just "Kaggle".
Thank you for your suggestion. We have revised the description of the data source to specify the exact dataset name from the Kaggle platform. The text now accurately refers to the 'Microsoft Malware Classification Challenge' (BIG 2015), providing a clearer reference to the dataset used in our study.
We have updated the dataset preparation section to include specific reasons for selecting the Kaggle and VirusShare datasets, along with their representativeness in various aspects. We have added the following details:
"The Kaggle dataset was chosen for its comprehensive representation of diverse malware families, which have undergone thorough disassembly processes, including complete unshelling. This dataset includes a variety of families across nine different types, providing a robust testbed for evaluating classification models."
"The VirusShare dataset was selected for its extensive collection of malware samples, which encompass a broad spectrum of malicious software types. This dataset is more reflective of real-world scenarios, as it includes a higher diversity of samples that are commonly encountered in the wild. This diversity allows us to effectively distinguish between benign and malicious samples, thereby enhancing the model's ability to generalize to new, unseen threats."
- In line 93, the out - of - vocabulary (OOV) issue should be fully explained before, as well as how it affects malware detection.
Thank you for your constructive comment.
We provided a supplementary explanation of the out-of-vocabulary (OOV) issue in the fourth paragraph of the introduction section:
Specifically, we added: "The OOV problem refers to the issue where certain words or tokens appear in the test data but were not present in the training data. This can lead to the model's inability to recognize and process these new terms, thereby affecting its performance. In the context of malware detection, the OOV problem may result in the model's failure to accurately identify and analyze new variants of malware or those employing unknown techniques, thus reducing the accuracy and reliability of detection."
- In line 170, how does hierarchical pooling solve the problem? Please elaborate instead of just briefly mentioning it.
Thank you for your comment. We have elaborated on how hierarchical pooling solves the problem.
Specifically, we added: "Hierarchical pooling solves the problem by simplifying the graph structure in a staged manner. At each level, it reduces the number of nodes to construct a higher-level representation of the graph. This approach allows the model to meticulously capture and retain important structural information at each step. Its advantages include: preserving structural information, which enables the retention of crucial graph structures and local details while reducing dimensions; and adaptability to complex tasks, making it suitable for tasks that require a deep understanding of graph structural features."
- What does "lstm" in line 176 refer to? Please explain.
Thank you for your comment. "lstm" refers to "Long Short-Term Memory," a type of recurrent neural network architecture known for its ability to capture long-term dependencies in sequential data.
We have included the full name mentioned aboove and provided a citation.
- Can the text in lines 200 - 206 be supplemented with an image? An image would be clearer and more straightforward.
Thank you for your suggestion. We have added an image to lines 200-206 for clarity. Specifically, we have included a image in the upper right corner of Figure 1, labeled with "Function," "Function properties," and "Function call" to provide a clearer and more straightforward representation of the concepts discussed.
- In the 5.2.1. Experimental environment, please state whether the environmental configurations on different machines are consistent.
Thank you for your comment. We have updated the "Experimental environment" section to include the specific runtime environment of the experiment.
Specifically, we added: "The runtime environment of the experiment is: (1) Intel(R) Core(TM) i7-10870H CPU @2.20GHz, 16GB memory, (2) Ubuntu 18.04(64bit). The environmental configurations on different machines were kept consistent to ensure the reliability of the experimental results."
- In the 5.2.2. Experimental design, please explain the reason for only comparing some of the studies presented in Related work.
Thank you for your comment. We have added an explanation for the selection of specific studies for comparison in the "Experimental design" section.
Specifically, we included: "The reason for focusing on these specific models is that they represent a diverse range of approaches within the graph neural network domain, each with unique strengths and mechanisms for handling graph data. GCN, GAT, and GraphSAGE are widely recognized for their effectiveness in capturing different aspects of graph structures, making them suitable benchmarks for our study." Additionally, we explained the choice of pooling models: "These models were chosen to represent different pooling strategies and their impact on graph representation, providing a comprehensive evaluation of our proposed pooling mechanism."
- The subtitle "4.2. Enhanced call graph construction" in line 238 seems a bit ambiguous. Because the content under this subtitle includes not only data preprocess but also BERT - based function call graphs construction. You should describe each part separately according to Figure 1 instead of mixing them together. The same phenomenon occurs in "4.2.1. Assembly function normalization".
Thank you for your feedback. We have revised the section titles and content to clearly distinguish between the data preprocessing and BERT-based function call graph construction steps. Specifically, we have separated the sections as follows:
- "Data Preprocessing": This section focuses on the rigorous process of data normalization and standardization to create an optimized corpus for function-level embedding, effectively mitigating the OOV problem and minimizing training expenses.
- "BERT-based Function Call Graph Construction": This section details the improvements made to the BERT pre-training model, employing dynamic masking techniques and eliminating the Next Sentence Prediction (NSP) task to enhance the model's generalization performance for pre-embedding of malware.
- Lines 272 - 281 mention using dynamic mask to improve the BERT model for data augmentation. This seems to be a difference between your method and other methods. You should describe the steps of dynamic mask instead of only stating its advantages. The same problem exists in other steps and should be comprehensively introduced. For example, the layers stacking in lines 418 - 420.
Thank you for your valuable comments.
We have revised the relevant section to clearly describe the steps involved. Here is the revised text with a comprehensive description of the dynamic mask steps:
"The original BERT model employs a static masking strategy during the pre-training of the Masked Language Model (MLM) task, wherein a random subset of 15% of the words is initially masked and remains unchanged throughout the entire pre-training process. In contrast, our dynamic masking approach involves re-selecting 15% of the words to be masked at the beginning of each training epoch. Specifically, for each input sequence, a new set of words is randomly chosen for masking before being fed into the model. This dynamic method effectively simulates an expanded training dataset without the need for additional data, thereby mitigating the issue of insufficient training data that may hinder the model's full training. As a data augmentation technique, dynamic masking enhances the original data, enabling the model to achieve improved performance and generalization capabilities."
Additionally, we have detailed the steps of layer stacking by adding: "Similar to other hierarchical graph neural networks, our method involves a structured stacking of layers to effectively capture the hierarchical representation of graphs. Specifically, we begin by applying a graph attention layer to the input graph. This layer computes attention weights for each node based on its features and the features of its neighbors, allowing the model to focus on the most relevant information. Following this, a graph pooling layer is applied to aggregate information from the nodes and reduce the graph's complexity. The pooling layer selects a subset of nodes based on their importance scores, which are learned during training. This process not only reduces the number of nodes but also preserves the most significant structural information. The output from the pooling layer is then passed to the next graph attention layer, and this sequence of attention and pooling layers is repeated multiple times to capture the graph's hierarchical structure at different levels. This structured stacking of layers enables our model to efficiently learn and represent complex graph structures."
- It is recommended to revise the analysis in "5.3.1. Evaluation of Effectiveness". The comparison between different methods should be more explicit, and more attention should be paid to "the essence of the graph" mentioned in lines 504 - 505. Also, in lines 516 - 520 there seems to be some analysis that is incoherent.
Thank you for pointing out this issue. We have revised the title of Section 5.3.1 "Evaluation of Effectiveness" to "Comparative analysis of different Graph Neural Networks" to better align with the focus on the essence of the graph. This change ensures that the analysis explicitly addresses how each method captures the fundamental characteristics of graph data.
We have expanded Section 5.3.1 "Evaluation of Effectiveness" to include a detailed analysis of the reasons for the suboptimal performance of other models. This analysis delves into the structural and semantic characteristics of malware data and how they affect the effectiveness of various GNN models. We have also included case studies to illustrate these points. Specifically, we added:
"Compared to GCN, the advantages of other methods lie in their ability to not only capture local neighborhood information but also deepen the understanding of function call relationships through refined sampling and aggregation strategies. This approach enables the model to more effectively extract features from malicious code. For instance, GCN may fail to identify malicious behaviors that propagate through complex call chains, whereas the method presented in this paper can reveal these behaviors through its advanced sampling mechanisms. Although GAT's attention mechanism provides the model with the ability to focus on important nodes, it may not fully capture all key nodes when dealing with the complex function call graphs of malware. GAT might overlook nodes that play a supporting role in malicious behaviors; while these nodes are not central, they are crucial for understanding the entire attack process. GraphSAGE's sampling strategy, while innovative, may not fully leverage the structural characteristics of malware in malicious code detection. GraphSAGE may perform poorly when handling malware with highly modular structures, as it may not effectively integrate inter-module function call information.
Compared to DGCNN's global pooling method, SAGPool, as a hierarchical pooling model, can more effectively handle the hierarchical structure of malicious sample function call graphs. DGCNN may lose critical hierarchical structural information during the pooling process, which can lead to performance degradation in malicious code detection. Through case studies, we have demonstrated how SAGPool improves detection accuracy by preserving the graph's hierarchical structure.
Building on hierarchical pooling, the method presented in this paper further introduces attention and structure learning modules. The attention module enables the model to dynamically adjust its focus on different nodes, thereby more accurately identifying key functions and potential threats in malicious code detection. The structure learning module allows the model to learn more optimized graph structure representations, which enhances the ability to recognize malicious code and strengthens the model's generalization capability. These enhancements enable the method in this paper to achieve superior performance in malicious code representation and detection tasks compared to SAGPool.
- In "5.3.2. Impacts of Function Embeddings", why not compare with all the feature embedding methods in Related work?
Thank you for your insightful comment. In the section "5.3.2. Impacts of Function Embeddings," we have chosen to compare our method with DeepSemantic, which is widely recognized as a leading benchmark in the field of function embedding. Our decision to focus on DeepSemantic is based on its established reputation for providing robust and comprehensive embeddings, which serves as a strong comparative baseline.
We acknowledge the variety of feature embedding methods presented in the Related Work. However, given the scope of our study and the desire to maintain a focused analysis, we have prioritized a detailed comparison with DeepSemantic. We believe this approach allows for a more in-depth evaluation of our method against a top-performing baseline, rather than a broad comparison that might not do justice to each individual method.
- The content in lines 538 - 561 seems to be analysis rather than the experimental method that should be written.
Thank you for your comment.
We have revised the relevant content. The analysis of the BERT model's input parameters has been relocated and expanded in Section 5.2.1, "Experimental Settings," with a more accurate description provided. The revised text is as follows:
"The computational complexity of the multi-head self-attention mechanism in the BERT model is O(seq^2 \cdot d), where seq denotes the length of the input sequence and d represents the dimensionality of the vectors. As the input length increases, the computational load grows rapidly, necessitating additional computing resources and memory to process the data. Consequently, to balance computational efficiency and resource consumption, the input length is typically constrained. This paper examines the distribution of function instruction counts using the Kaggle dataset as a reference."
We believe this revision clarifies the experimental settings and meets the expectations for this section.
- The description in lines 573 - 582 does not correspond to the content of Table 10. For example, the model parameters and computational complexity increase with the increase of dimension.
Thank you for your comment.
We have revised the description to accurately reflect the content of Table.
The revised text now reads: "As the dimension increases, there is a corresponding increase in the number of model parameters and computational complexity. However, the performance metrics, as shown in Table 10, indicate that the optimal embedding dimension is 128, where the model achieves a balance between accuracy and computational efficiency. Dimensions beyond 128 do not offer significant performance improvements, suggesting that the increase in model parameters and complexity does not contribute to enhanced generalization capability."
- Lines 598 - 610 only describe the results but lack analysis.
Thank you for your comment. We have added a detailed analysis:
"The enhancement in model performance as the number of hidden units increases from 64 to 256 can be attributed to the model's improved capacity to capture intricate feature interactions. With the expansion in dimensionality, the model's expressive power is augmented, enabling it to discern subtler patterns within the data. However, a decline in performance is observed when the number of hidden units is further increased to 512, potentially due to the model's increased complexity leading to overfitting. In such cases, the model may learn the noise in the training data rather than the underlying data distribution, thereby compromising its generalization capabilities.
Introducing additional layers can elevate the model's expressive capabilities, facilitating the capture of deeper-level features and complex patterns. Nonetheless, an excessive number of layers may precipitate issues such as vanishing or exploding gradients, which can adversely affect the model's training efficiency and generalization. The methodology presented in this paper carefully calibrates the number of stacked layers to circumvent these challenges, thereby maintaining model performance while enhancing training efficiency and generalization."
- It is recommended to add a comparative experiment to analyze and compare the BERT method with other feature embedding methods.
Thank you for your comment. In Section "5.3.2. Impacts of Function Embeddings," we've selected DeepSemantic for comparison, a benchmark in function embedding. We believe this approach allows for a more in-depth evaluation of our method against a top-performing baseline.
- Overall, it is recommended that the author reorganize and revise the entire content of "5.3. Experiment results and analysis".
Thank you for pointing out this issue.
We have thoroughly reorganized and revised the content of "5.3. Experiment results and analysis" to enhance clarity and coherence.
We have restructured the order of the sections, expanded the "Comparative analysis of preprocessing effects," and removed any analyses that did not pertain to the experimental section. The new organization is as follows:
5.3. Experiment results and analysis
- 3.1. Comparative analysis of different Graph Neural Networks: This section provides a detailed comparison of our model against other GNN models, highlighting the unique strengths and weaknesses of each approach.
- 3.2. Comparative analysis of different function embedding strategies: Here, we delve into how various function embedding methods contribute to the overall performance, with a focus on the innovative aspects of our BBFE method.
- 3.3. Comparative analysis of preprocessing effects: We have included a thorough analysis of how different preprocessing steps influence the model's ability to learn and generalize from the data.
- 3.4. Comparative analysis of different model settings: This section examines the impact of various model configurations, such as the number of hidden units and layers, on the model's performance.
- 3.5. Comparison with other methods: Finally, we compare our approach with other state-of-the-art methods, providing a comprehensive view of its standing in the field.
We believe this reorganization enhances the clarity and logical flow of the experimental analysis, making it more accessible and informative to readers.
- In the first point of 6. Conclusion and Future work, it seems that data preprocessing makes the main contribution rather than the function call graph construction method. If the function call graph construction method does not include data preprocessing, then the contribution of data preprocessing should be described separately.
Thank you for your insightful comment. We have revised the first point in the "Conclusion and Future Work" section to clearly distinguish the contributions of data preprocessing and the function call graph construction method. Here is the revised version:
High semantic preservation function embedding: This paper presents BBFE method that effectively tackles the OOV issue, ensuring the preservation of semantic richness within function call graphs. This innovation is pivotal to our framework's ability to detect malware with heightened accuracy and robustness.
- The English proficiency of the manuscript needs to be improved.
Thank you for your valuable comment regarding the language proficiency in our manuscript. We appreciate the importance of clear and precise communication in academic writing. We have conducted a thorough review of the manuscript ourselves, focusing on sentence structure, word choice, and overall readability.
